# communications
# engineering

# Large area optimization of meta-lens via data-free machine learning

Maksym Zhelyeznyakov [1✉], Johannes Fröch[1,2], Anna Wirth-Singh[2], Jaebum Noh[3], Junsuk Rho [3,4,5], Steve Brunton [6] & Arka Majumdar [1,2✉]

Sub-wavelength diffractive optics, commonly known as meta-optics, present a complex numerical simulation challenge, due to their multi-scale nature. The behavior of constituent sub-wavelength scatterers, or meta-atoms, needs to be modeled by full-wave electro-magnetic simulations, whereas the whole meta-optical system can be modeled using ray/Fourier optics. Most simulation techniques for large-scale meta-optics rely on the local phase approximation (LPA), where the coupling between dissimilar meta-atoms is neglected. Here we introduce a physics-informed neural network, coupled with the overlapping boundary method, which can efficiently model the meta-optics while still incorporating all of the coupling between meta-atoms. We demonstrate the efficacy of our technique by designing 1mm aperture cylindrical meta-lenses exhibiting higher efficiency than the ones designed under LPA. We experimentally validated the maximum intensity improvement (up to 53%) of the inverse-designed meta-lens. Our reported method can design large aperture ($\sim 10^4 - 10^5\lambda$) meta-optics in a reasonable time (approximately 15 minutes on a graphics processing unit) without relying on the LPA.

[1] Department of Electrical and Computer Engineering, University of Washington, Seattle 98195 Washington, USA. [2] Department of Physics, University of Washington, Seattle 98195 WA, USA. [3] Department of Mechanical Engineering, Pohang University of Science and Technology (POSTECH), Pohang 37673, Republic of Korea. [4] Department of Chemical Engineering, Pohang University of Science and Technology (POSTECH), Pohang 37673, Republic of Korea. [5] POSCO-POSTECH-RIST Convergence Research Center for Flat Optics and Metaphotonics, Pohang University of Science and Technology (POSTECH), Pohang 37673, Republic of Korea. [6] Department of Mechanical Engineering, University of Washington, Seattle 98195 WA, USA. ✉email: mzhelyez@gmail.com; arka@uw.edu

n the age of silicon computing, numerical simulations are at the heart of understanding and designing physical systems. For many cases, analytical solutions to complex device geometries are intractable to compute, or simply do not exist. From extremely large systems like rockets[1] to ultra-small nanophotonic devices[2], numerical simulations provide scientists and engineers with the necessary tools to design nonintuitive structures. In electromagnetics, direct solvers, including the finite difference time domain (FDTD)[3] and the finite difference frequency domain (FDFD)[4,5] simulators, are the usual choices when dealing with heterogeneous structures with subwavelength features that require a high degree of numerical accuracy. Most commonly, electromagnetic simulation tools serve to validate the qualitative designs created by engineers based on prior knowledge and intuition. In recent years, the field of nanophotonics has incorporated a new paradigm of computer-aided device design, where a device's performance is summarized by a quantitative figure of merit (FOM) that is optimized over. This method involves running a forward numerical simulation, computing the FOM, and iteratively modifying the device's geometry based on an optimization algorithm to reach the desired FOM. Such optimization methods, often termed as inverse design, have already been used to create multi-functional and efficient nanophotonic structures[2,6–15]. However, electromagnetic simulators suffer from a computational resource problem when the device dimension becomes large ($\gtrsim 10^3\lambda$), where $\lambda$ is the device's operating wavelength. As most electromagnetic simulations are performed over a sub-wavelength grid size, with increased size, the number of input variable becomes prohibitively large, making the simulation slow and memory extensive. The limitation of such forward electromagnetic simulators becomes even more severe for inverse design, where many such forward simulations are needed.

Sub-wavelength diffractive optics, also known as meta-optics, present an important test-bed for these problems: the constituent elements of the meta-optics, i.e. meta-atoms, are sub-wavelength, but the dimensions of the whole meta-optics are on the order of $\sim 10^3\lambda - 10^5\lambda$. Thus the underlying physics of each scatterer has to be modelled using full-wave electromagnetic simulation, but the whole meta-optical system needs to be simulated using ray or wave optics. Such multiscale electromagnetic simulators invariably rely on approximations, the most common of which is the local phase approximation (LPA): the scattering in any small region is taken to be the same as the scattering from a periodic surface[9]. This approximation allows the simulation of each scatterer in a periodic array, abstracting out the electromagnetic response as a simple phase shift. While this significantly reduces the computational complexity of simulating a meta-optic, this approximation fails to consider the coupling of each scatterer with their dissimilar neighbors. In fact, it has already been shown that meta-optical lenses designed under LPA have suboptimal efficiencies[16], especially when the numerical aperture is large. The LPA becomes even more inaccurate when the material used to create the meta-lens has low index, such as SiN[17]. We note that, while a full FDTD coupled with adjoint optimization has been used to design a meta-optic without relying on LPA, their size has been limited to only $\sim 100\lambda$[18]. LPA can also be bypassed using Mie scattering approaches[19], which however limits the shape of scatterers.

To address the computational bottleneck of large-area inverse design, here we introduce a physics-informed neural network (PINN), to model super-cell subsections of a larger metasurface[20–22] which in conjunction with the overlapping boundary method[23–27], can replace a traditional FDTD/ FDFD solver to predict the electric field distribution for a given dielectric distribution. PINNs and other model-based deep learning architectures have already been used in modeling physical systems[28].

We also note that a large number of works already used artificial neural networks to predict spectral responses of meta-optics of varying scatterer geometries[25,29–36]. However, these works used largely periodic structures for which LPA is accurate. We present a solution via PINNs[37,38] for lenses and devices with spatially varying scatterer geometries, where it is necessary to model the whole electric field from several scatterers and their neighbors. The use of PINNs to accurately model the electromagnetic scattering beyond the LPA is the main contribution of this work. PINNs solve partial differential equations (PDEs) by minimizing a loss function constructed from the PDE itself. This loss function is generally some norm of the residual[37] or an energy function derived from the PDE[39]. PINNs have already seen wide usage in the field of fluid mechanics[40–42], biology[43], and solving stochastic PDEs[44]. In electromagnetic inverse problems, PINNs have also been employed to design meta-optics and nanophotonic devices[45,46]. These works, however, did not clearly demonstrate a simulation speedup, and are limited to the inverse design of only very small devices. We also note that pre-trained PINNs have been used to design small gratings[47]; however their methodology is limited to small gratings that deflect light fields to specific angles, and thus cannot be readily used for the inverse design of arbitrary meta-optics or a meta-lens.

In our work, we train PINNs to predict the electric fields from a parameterized set of dielectric meta-atoms corresponding to rectangular pillars. We then use this as a surrogate model to design cylindrical meta-lenses operating in the visible with a diameter of 1 mm ($\sim 1500\lambda$). Large area meta-optics are simulated by partitioning the simulation region into groups of 11 meta-atoms, with the outermost meta-atoms overlapping. After simulation, the fields are stitched together. Our PINNs do not require a training data set. They are trained by randomly generating distributions of dielectric meta-atoms $\epsilon$, feeding them into a neural network $NN$, and minimizing the residual of the linear Maxwell PDE operator

$$\left\| A_{\mathrm{Maxwell}}(\epsilon)NN(\epsilon) - b \right\|_1 \qquad (1)$$

over the neural network training parameters. This means our PINNs are trained without ever invoking a forward numerical simulation of Maxwell's equations during the training process. Numerical simulations are invoked only to test the neural network performance (see next section, Supplementary Note 6, and Supplementary Fig. 5). A similar data-free approach has been applied to deep-tissue microscopy[48], however inverse design was not demonstrated. Once trained, this method can calculate the full electromagnetic field response from a 1 mm diameter cylindrical meta-lens at $\sim 630$nm in approximately 3 seconds on a graphics processing unit (GPU). Furthermore, we demonstrate an experimental improvement (over 50%) of the maximum intensity of cylindrical metalenses over their forward designed hyperboloid counterparts, signifying the improvement over using LPA. We note that the reported method is robust enough to handle even larger meta-optics, with simulation time scaling only linearly with the aperture of the cylindrical lens (see Supplementary Note 9).

## Methods
**Deep neural network proxy to Maxwell's equations.** Our problem statement is summarized in Fig. 1c. The monochromatic electromagnetic scattering equation for an inhomogeneous, nonmagnetic material is given by:

$$\nabla \times \nabla \times \mathcal{E}(x) - \omega^2 \epsilon(x)\mathcal{E}(x) = i\omega\mathcal{J}(x). \qquad (2)$$

In the 2D case, assuming out of plane polarization $(0, 0, \mathcal{E}_z)$, and the double curl vector identity, $\nabla \times \nabla \times = \nabla ( \nabla \cdot ) - \nabla^2$ we can

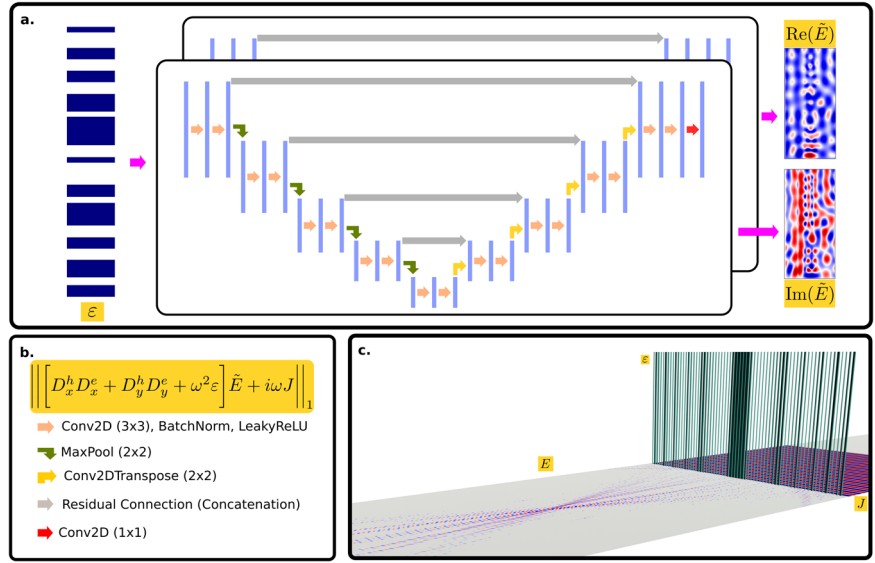

**Fig. 1 Problem outline. a** Neural network schematic. (For a more detailed schematic see Supplementary Fig. 10). $\varepsilon$ distributions of 11 pillar meta-optics are meshed by randomly generating sets of pillar half-widths of height $h = 0.6\mu m$ with a dielectric constant 4 corresponding to SiN. The background medium is air. The loss function is the $\| \cdot \|_1$ norm of the residual of Eq. (2). **b** Neural network architecture. Encoder layers are down-sampled by a maxpool operation with a 2 × 2 kernel. The decoder part of the network is up-sampled by the Conv2DTranspose operation with a 2 × 2 kernel. **c** Render of the system under optimization. A current $J$ is incident on a cylindrical metalens with dielectric distribution $\varepsilon$, with output response $E$.

simplify Eq. (2) to:

$$\nabla^2 \mathcal{E}_z(x) + \omega^2 \epsilon(x) \mathcal{E}_z(x) = -i\omega \mathcal{J}_z \quad (3)$$

where $\mathcal{E}_z$ and $\mathcal{J}_z$ are scalar fields. Equation (3) is defined over all space, with boundary conditions at $|x| \to \infty$. To simulate this equation, we discretize it on a Yee grid[3] by replacing the $\nabla$ operator with a matrix, and treating the field $\mathcal{E}_z(x)$ and current $\mathcal{J}_z$ as vectors $E$ and $J$ at discrete values of $x$. Similarly, we treat the dielectric distribution $\epsilon(x)$ as a diagonal matrix $\varepsilon$. To truncate the simulation to a finite domain, we use perfectly matched boundary layers (PML), by making the transformation on the partial derivative operators $\frac{\partial}{\partial x} \to \frac{1}{1+i\frac{\sigma(x)}{\omega}}\frac{\partial}{\partial x}$. Making these substitutions, Eq. (3) becomes:

$$\left[ D_x^h D_x^e + D_y^h D_y^e + \omega^2 \varepsilon \right] E = -i\omega J \quad (4)$$

with matrices $D_x^h, D_x^e, D_y^h, D_y^e$ being the matrix representations of corresponding derivative operators on a Yee grid with incorporated PML boundaries. See Supplementary Note 5 and Supplementary Fig. 4 for a more detailed description of the matrices. These matrices were extracted from a modified version of the package angler[49] with constants $c, \epsilon, \mu$ set to 1 and the length scale set to μm. To build a neural network proxy to solve Eq. (4), we employ a PINN (Fig. 1a and b). PINNs generally use the coordinates of the computational grid as the input to the neural network, and then minimize the residual of the physical equations by approximating the target quantity being solved for with a neural network. This approach is slow since it effectively functions as an iterative solver re-parameterized over neural network weights and biases. It also required retraining the neural network for all different dielectric distributions. Our approach is to build a proxy solver that predicts the field $E$ from a dielectric distribution $\varepsilon$. We pretrain the PINN to predict fields from inputs $\varepsilon$ before optimizing our meta-lenses. The minimization problem to train

the PINN becomes:

$$\min_\theta f(\varepsilon; \theta)$$

$$\text{where} \quad f(\varepsilon; \theta) = \left\| \left[ D_x^h D_x^e + D_y^h D_y^e + \omega^2 \varepsilon \right] NN(\varepsilon; \theta) + i\omega J \right\|_1 \quad (5)$$

with $NN(\varepsilon; \theta)$ being the output field from the PINN, and $\| \cdot \|_1$ is the vector $l_1$ norm. Here $\theta$ refers to the weights and biases of the neural network $NN$. A lower physics informed loss indicates that the neural network is actually satisfying the PDE, and thus predicting the field more accurately. We re-emphasize that there is no data term in $f(\varepsilon; \theta)$, which simplifies the neural network training process. Furthermore, we believe that it mitigates the accumulation of error in the gradients during the inverse design process observed by Chen et. al.[47]. Figure 1 outlines the general strategy for building the proxy model. During each epoch, 10 (batch size) dielectric distributions consisting of rectangular pillars of height $h = 0.6$ μm with dielectric constant 4 (corresponding to SiN), are generated from 11 random pillar half-widths per batch. The operation wavelength is $\lambda = 0.633$ μm. The neural network architecture chosen is a UNET, shown in Fig. 1a and **b**, due to previously reported good performance with scattering problems[47]. The model is trained for $5 \times 10^5$ epochs using the ADAM optimizer[50] with a learning rate set to $5 \times 10^{-4}$. The final residual of the fields predicted by the neural network are of the order of ~0.5, compared to the numerical residual produced by FDFD which is on the order of $10^{-16}$. Although there is a large difference, in the next section we show that this still produces a simulator which is capable of outperforming the LPA when optimizing the efficiency of a metalens. Figure 2**a** shows an example of a field predicted from a random set of pillars by the neural network, by a 2D FDFD code, and their difference, showing good qualitative match. A more quantitative measure of the errors is shown in Fig. 2**b**, where we show the point-wise error probability density functions for the relative error between the complex fields predicted by FDFD and that predicted by the neural network and the field predicted under LPA, and the absolute error between pillar-wise average transmission

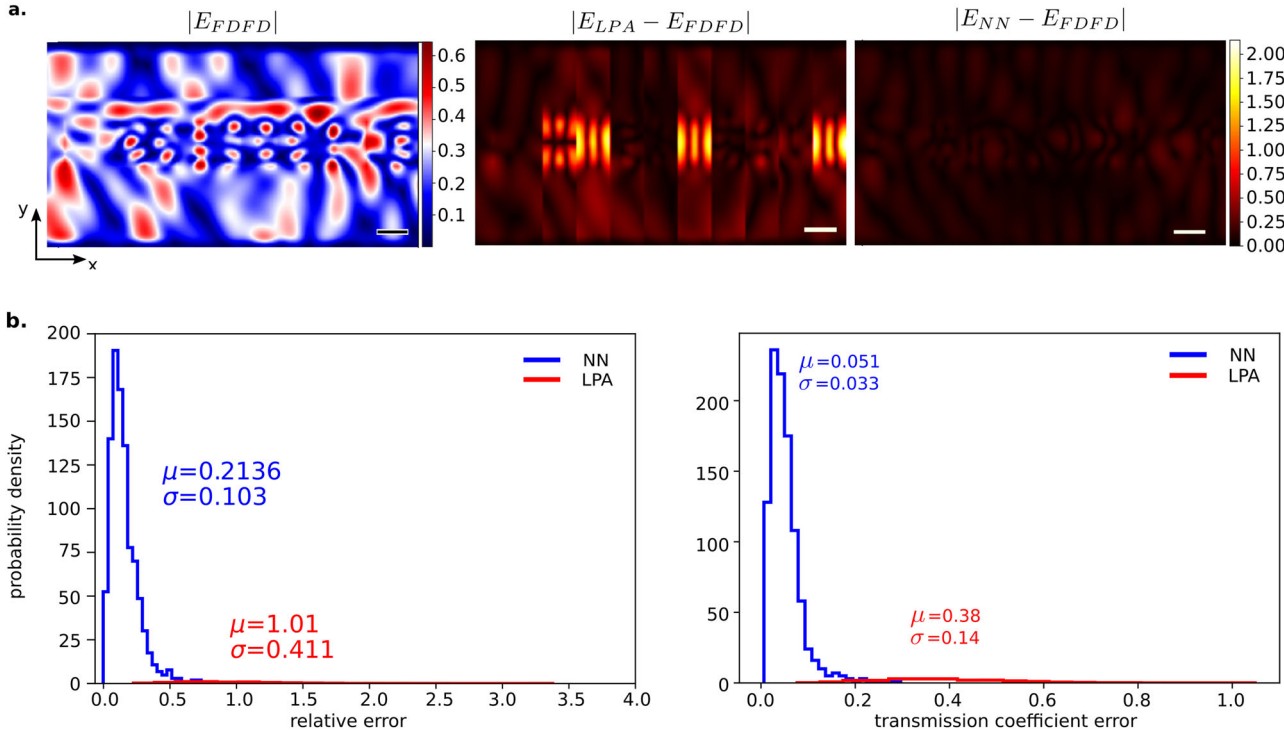

**Fig. 2 Neural network performance analysis. a** An example field. (left) is the absolute value of the ground truth finite difference frequency domain (FDFD) simulated field, (middle) difference between the true value of the field and the field produced by using the local phase approximation (LPA), and (right) is the absolute difference between the neural network field and the FDFD field. **b** Comparison between the performance of the proposed neural network and LPA methods. (left) Shows the relative error between FDFD predicted fields and the fields predicted by LPA. (right) Error comparisons between the transmission coefficients predicted by LPA and the neural net.

coefficients. See Supplementary Note 3 for a more detailed description of the pillar wise transmission coefficient error. The relative error is expressed as:

$$\frac{\left\| E_{\text{approx}} - E_{\text{FDFD}} \right\|_2^2}{\left\| E_{\text{FDFD}} \right\|_2^2}. \tag{6}$$

For the PINN, $E_{\text{approx}}$ is the field predicted from a set of 11 pillars. For the LPA, $E_{\text{approx}}$ is fields predicted from the same set of pillars, and then stitched together over the same region. See Supplementary Fig. 2 for a visual explanation. The mean expected relative error for the neural network is $\mu = 0.21$ with a standard deviation of $\sigma = 0.103$. When using the LPA over the same region, we get a mean relative error of $\mu = 1.01$ with a standard deviation of 0.411. Thus, based on the relative field error, our method is $4.8 \times$ more accurate than the LPA. For the pillar-wise transmission coefficient error, we get an expected error of $\mu = 0.051$ for the neural network with a standard deviation of $\sigma = 0.033$ and for the LPA method we get an expected error of $\mu = 0.38$ with a standard deviation of 0.14. Thus, based on the transmission coefficient error, our method is $7.2 \times$ more accurate than the LPA.

**Device optimization**. The optimization process based on automatic differentiation functionality of PyTorch for large area meta-optics is outlined in Fig. 3. The forward problem is solved via a pre-trained PINN. Since the input into the neural net is a meshed grid of pillars, a differentiable map from pillar half-widths (3a.) to meshed geometries (3b) must be generated. This is achieved by generating Gaussian functions centered around pillar centers, with standard deviations of pillar half-widths in the x dimension, and pillar height in the y dimension, and then using a modified softmax function to transform the Gaussians into rectangles with slightly rounded edges, making them differentiable via automatic differentiation (see Supplementary Note 4 and Supplementary Fig. 3). The meshed structures are fed into two separate neural networks that have been pre-trained to predict the complex electric field (3c.). The fields are then stitched together with regions of the outer half-widths overlapping. The total field is then propagated using the angular spectrum method (3d). The propagated field is used to calculate the FOM $f$ (3e.) from Eq. (5). We use automatic differentiation to compute the gradients of the FOM with respect to the input half-widths $\nabla_{\vec{r}} f$, and iteratively update them with the ADAM optimizer[50].

## Results

We used the PINN surrogate model to optimize 9 different lenses, all with 1 mm aperture, with focal lengths ranging from 250 to 1500 μm in increments of 250 μm. The minimum feature size is set to 75 *nm*, to ensure fabricability. To compare our optimization approach, we also generated lenses according to the hyperboloid phase equation:

$$\phi(x, y) = \frac{2\pi}{\lambda}\left(\sqrt{x^2 + F^2} - F\right) \tag{7}$$

The phase is implemented under LPA using SiN (refractive index 2), a wavelength of 0.633 μm, and periodicity of $p = 0.443$ μm (see Supplementary Fig. 9). We then optimize the lens employing our PINN to increase the intensity at the focal spot, i.e., the FOM is given by:

$$f = |E(x = 0, z = F)|^2 \tag{8}$$

Figure 4**a** and **b** show the intensity profile of a forward designed and optimized lens with $F = 500$ μm focal length. Figure 4**c** shows the normalized intensity slice at the focal spot of both lenses. As

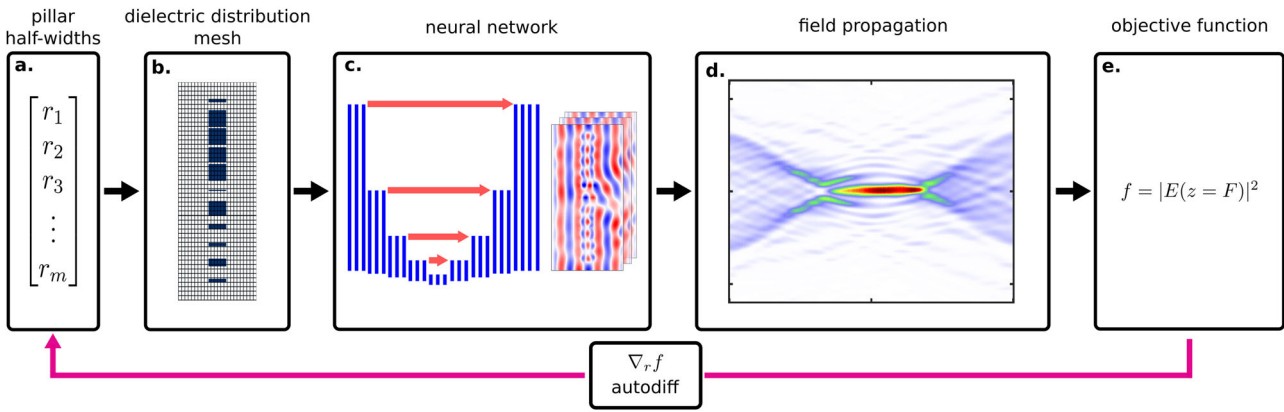

**Fig. 3 Optimization strategy of 2D meta-optics with physics informed neural networks (PINNs). a** We start with a vector, which contains a list of all pillar half-widths, characterizing the meta-optic. These half-widths are then batched into groups of 11 with an overlap of 1 pillar on each side (see Supplementary Note 2 and Supplementary Fig. 1 for notes on the computational domain set up). The choice of 11 pillars was made based on the GPU memory required to train the PINN. **b** The half-widths are meshed into dielectric distributions which get fed into the neural network. **c** The neural network predicts patches of fields which are then stitched together, and **d** propagated via the angular spectrum method. **e** The objective function is formed from the resulting field, and backpropagated using PyTorch's automatic differentiation functionality to update the initial radius distribution.

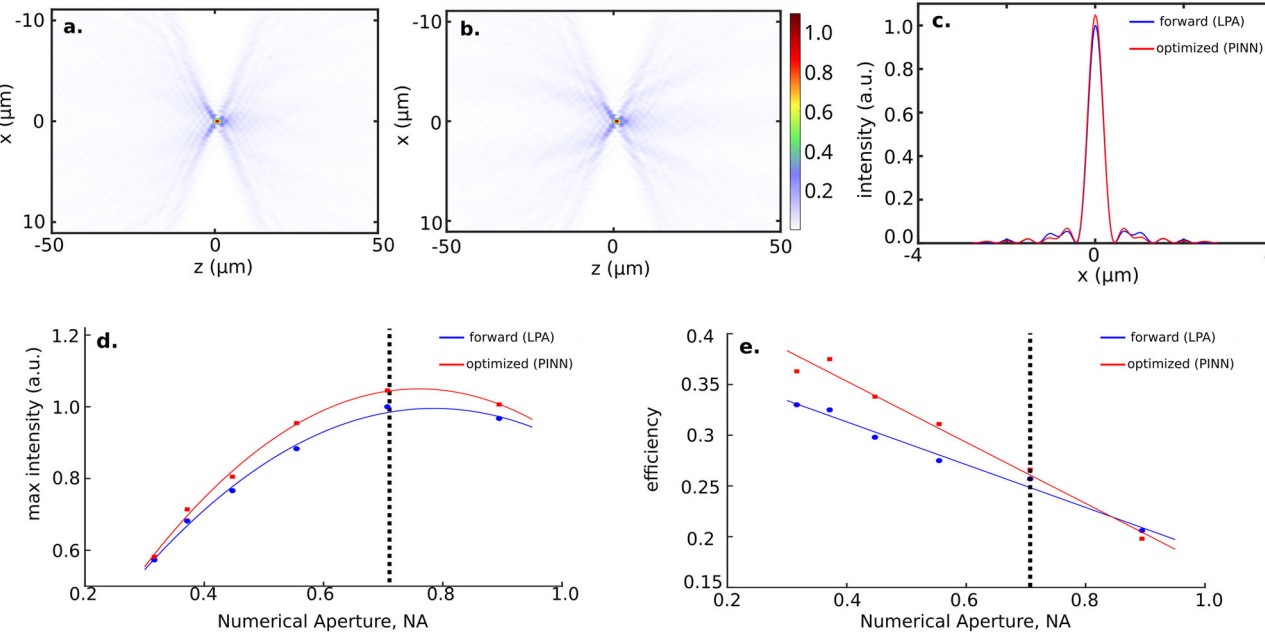

**Fig. 4 Efficiency and intensity sweeps of forward designed lenses and optimized lenses. a** Focal spot intensity profile of a forward designed lens with focal length $F = 500\mu m$. **b** Focal spot intensity profile of an optimized lens. **c** Slices of intensity profiles for both lenses. The intensity was normalized such that the maximum intensity of the forward designed lens is 1. The theoretical performance improvement is ~3%. **d** Maximum intensity at the focal spot vs lens numerical aperture (NA). Intensities are normalized such that the maximum of the largest forward designed intensity is set to 1. **e** Theoretically computed efficiencies of the lenses vs NA. The solid lines are visual aids for the trend and do not correspond to a theoretical prediction.

seen in Fig. 4**d** the maximum intensities at the focal spots improve in every case. Figure 4**e** shows that the efficiency improves in all except for the lens with the highest NA. We also find a trend that the improvement in the maximum intensity of the inverse-designed meta-lens over the forward-design meta-lens increases with increasing NA. As with higher NA, the phase gradient becomes larger, we expect the LPA to be a worse approximation. Interpreting the efficiency improvement is more convoluted. We defined the efficiency as the ratio of the light energy inside a circle of radius of three times the full width half maximum (FWHM) at the focal spot over the total energy in the focal plane. With increasing NA, the FWHM decreases, making the efficiency improvement lower with increased NA. For the highest NA, the FWHM of the inverse-designed meta-lens is

significantly lower than the forward-designed meta-lens, making the efficiency lower.

We validated our designs by fabricating and experimentally testing the meta-lenses using a microscope (details of fabrication and characterization in Supplementary Note 1.1 and Supplementary Note 1.2). Figure 5 shows an example of the inverse optimized device. Figure 5**a–c** shows the scanning electron micrographs (SEMs) of the fabricated optimized lens with focal length $F = 500\mu m$. Figure 5**d** shows the distribution of the dielectric pillar half-widths of the same forward and optimized lens. signifying the two designs are very different. Figure 5**e** shows the focal spot intensities of the lenses integrated over a $r = 3 \times$ FWHM region at the focal spot, which yields a quantitative value to compare the lens efficiency[51] among different devices.

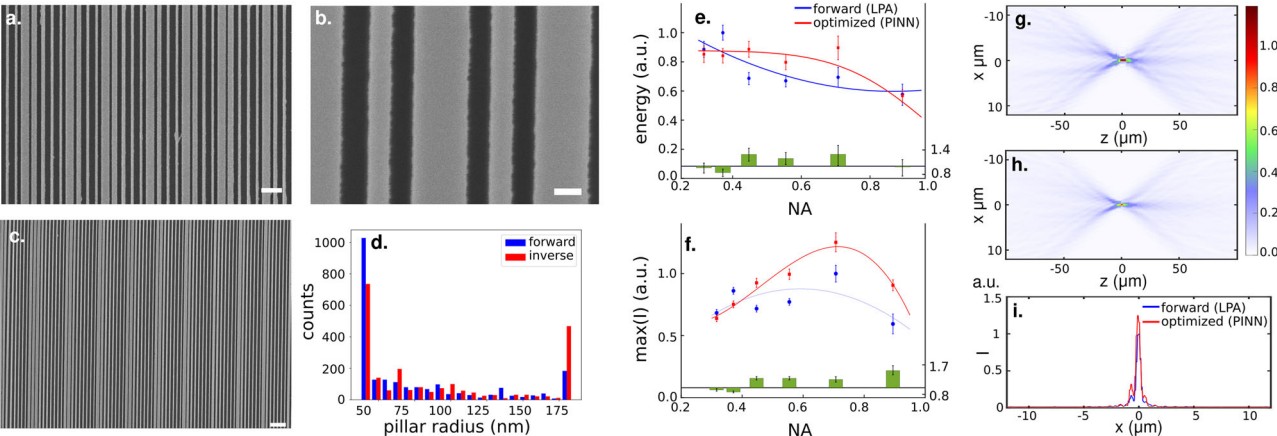

**Fig. 5 Experimental results. a–c** Scanning Electron Microscope (SEM) images of the fabricated SiN meta-lens with focal length $F = 500\mu m$. The scale bars correspond to 1 μm, 0.1 μm, and 1 μm, respectively. **d** Counts of pillar half-widths of the forward and inverse designed lens. **e** Measured intensity contained in the region given by 3 × FWHM of the focal spot vs lens numerical aperture. The units are normalized to the largest intensity integral of the forward design. Error bars are standard deviations of energy computed at 1216 data points (dimension of the sensor). **f** Maximum intensity at the focal spot. The inverse designed lenses outperform the forward designed lenses for NA> 0.44. The lines are visual aids and not fits to a theoretical model. Units are normalized to the largest intensity of the forward designed lens. In the NA = 0.9 ($F = 250\mu m$) case an improvement of 53% is observed. The error bars are standard deviations of intensity at the focal plane. **g** Experimentally measured field intensities of the forward designed lens and **h** of the inverse designed lens. **i** Intensity slice at the focal spot. The intensities are normalized such that the maximum intensity of the forward designed lens is 1. The intensity of the inverse designed lens focal spot is 1.25.

Figure 5**f** plots the maximum intensity plot as a function of the lens NA. For optimized lenses with NA > 0.44, we see improvements of more than 25%, with a maximum improvement of 53% for the NA = 0.9 lens. The experimentally determined intensity integral, which is analogous to the efficiency of a lens, on has improvements of more than 18% in all cases except for the NA=0.9 case. This is because the FWHM of the optimized lens at the NA=0.9 case is actually smaller than the FWHM of the forward designed lens, leading to a smaller integration area when computing the energy. We note that, while a quantitative match between the experiment and design is not obtained, we did observe a similar trend in terms of improved intensity and efficiency as predicted by the theory. Fig. 5**g** shows experimentally measured field profiles of the forward designed $F = 500\mu m$ meta-lens. Figure 5**h** shows the same for an optimized lens. Figure 5**i** is the slice of the focal spot intensity profile along the $z = F$ plane. In all these figures, the intensity is normalized such that the maximum intensity of the forward designed lens is 1.

## Discussion

We have developed a PINN to use as a proxy surrogate model for simulating the full Maxwell's equations to design dielectric meta-optics. We used the PINN to optimize pillar half-widths to maximize the intensity at the focal spot of 1 mm aperture cylindrical meta-lenses at 633nm. We demonstrated experimental improvements of the maximum intensity of the lenses up to 53%. We also want to note that this method was useful for the inverse design of extended depth of focus lenses[10] (see Supplementary Note 7 and Supplementary Fig. 6). This model did not use the LPA, but simulated meta-atoms by splitting up the device into chunks with overlapping boundaries, and stitching the chunks together to approximate the full field response. We emphasize that FDFD simulations were never carried out to train the PINN, and we only minimized the residual of the PDE itself to train the network. The PINN training took approximately 2 hours on our machine. In our studies, this method provided approximately a 3-5x speedup (see Supplementary Figure 8 and Supplementary Table 1) over conventional FDFD with overlapping boundary conditions, and was

much simpler to use as a forward simulator for optimization problems since it can be used as a simple map from $\epsilon$ to $E$-field with gradients computed by automatic differentiation.

We would also like to note that the theoretical intensity and efficiency improvements are smaller than their experimental counterparts. While we do not have a clear explanation for this discrepancy, the theoretical and experimental trends in lens improvement are similar. One hypothesis could be that the the inverse designed lenses may be more tolerant to fabrication imperfections. However, randomly changing the scatterers in our meta-optics by 10% did not give a similar enhancement. As such this aspect of the quantitative match between experiment and theory remains an open problem and further studies will be needed. We would like to point out however the importance of experimentally verifying inverse design methodologies, since in our studies we used open source codes that produce reasonable looking results, but are not experimentally accurate (see Supplementary Note 8 and Supplementary Fig. 7). The inverse design solution we introduced in this paper can be integrated into various computationally intensive tasks which require mate-optical inverse design such as the end-to-end optimization of computational imaging systems and the design of optical neural networks[52,53]. It is worth noting, however, that this method is not a general numerical solver. It is limited to predicting electromagnetic field responses from fixed source, material, and boundary parameters. Source type and $k$-vector, dielectric constant, geometry type (rectangular pillar of fixed height), and boundary conditions must all remain constant for this method to work. If any of these parameters are modified, the PINN must be retrained. Furthermore, the method we presented was only implemented under a 2D approximation. Extending this method to 3 dimensions would take significant effort due to the fact that the electric field $\mathcal{E}$ could no longer be treated as a scalar field, and the full vector nature would have to be modeled. On a $n \times n$ grid in 2D, the Maxwell operator $[\nabla^2 + \omega^2\epsilon]$ results in a $n^2 \times n^2$ matrix, while for a $n \times n \times n$ 3D grid the Maxwell operator $[\nabla \times \nabla \times - \omega^2\epsilon]$ result in $9n^3 \times 9n^3$ square matrices due to the additional 2 vector field components that must be modeled. However, these operators are sparse with a small number of

nonzero elements that scale as $\sim 38n^3$ in 3D, making small problems still manageable. The other problem with generalizing this method to 3D is the large null-space of the $\nabla \times \nabla \times$ operator which results in slow convergence of numerical methods[54,55]. It is highly likely that this could also affect the training of the PINN, and require regularization or preconditioning which deflates the null space of this operator to properly converge onto a solution. On the other hand, in this work we showed that machine-precision numerical accuracy of numerical solvers may be not be needed for inverse design methods with FDFD. Solvers could be sped up by relaxing the relative error tolerance, such that iterative solvers converge quicker for predicting the forward and adjoint problems. Another interesting aspect will be to understand the optimal PINN to model the meta-optics, and if we can identify a relationship between the number of trainable parameters and size of the problem we are solving. In future work we aim to explore these options.

## Data availability

Data sets generated in the paper are available from the corresponding author on reasonable request.

## Code availability

Code for the project is available at https://github.com/demroz/pinn-ms.

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

## Acknowledgements

This research was funded by NSF-GCR-2120774. M.V. Z. is supported by an NSF graduate fellowship. Part of this work was conducted at the Washington Nanofabrication Facility/Molecular Analysis Facility, a National Nanotechnology Coordinated Infrastructure (NNCI) site at the University of Washington, with partial support from the National Science Foundation via Awards NNCI-1542101 and NNCI-2025489.

## Author contributions

M.V.Z. conceptualized the project. M.V.Z. designed the methodology, wrote the software, trained the machine learning model, and performed the data analysis. J.F. manufactured the metasurfaces and experimentally validated them. A.W.S. validated metasurface designs in simulation. J.N. assisted with validating metasurface designs in simulation. J.R. provided compute resources for training the machine learning model. S.B. and A.M. supervised the project. Initial version of manuscript was written by M.V.Z.

## Competing interests

The authors declare no competing interests.

## Additional information

**Peer review information** : *Communications Engineering* thanks the anonymous reviewers for their contribution to the peer review of this work. Primary Handling Editors: Rosamund Daw, Mengying Su. A peer review file is available.

