## [Peer Review File · Communications Engineering]

Note: *This manuscript has been previously reviewed at another Nature Portfolio journal. This document only contains reviewer comments and rebuttal letters for versions considered at Communications Engineering.*

Reviewers' comments:

Reviewer #1 (Remarks to the Author):

In this revised version of the manuscript, the authors have barely addressed my main comments (apart from small typos fixes, adding references, etc.):

- they did not provide a comparison to other high-efficiency designs (nor did they show additional capability, which they claim is not interesting)
- they did not provide intuition for which their experiment sometimes outperforms experiments, and even indicate that their code has errors? I am not sure how to interpret that last part and the potential impact on the validity/reproducibility of the results?

Overall I do not believe that the results deserve publication in their current forms, and still encourage the authors to make significant improvements to the presentation and discussion of their results.

Reviewer #2 (Remarks to the Author):

The authors have addressed my comments. I would recommend publication in Communications Engineering.

Reviewer #3 (Remarks to the Author):

I believe the authors make an overall compelling case for publication in this journal, and that the revision has improved the work, especially in clarification of how it fits into and adds to the previous literature. I recommend publication as is.

In this revised version of the manuscript, the authors have barely addressed my main comments (apart from small typos fixes, adding references, etc.):

- they did not provide a comparison to other high-efficiency designs (nor did they show additional capability, which they claim is not interesting)

Response: We respectfully disagree with the reviewer. We did not claim that demonstrating additional capability is not interesting. We simply stated that the reviewer's statement "I believe that the field of metasurfaces is past the search for high-efficiency monochromatic metasurfaces" is untrue. We are working on several commercial and defense related projects, and "low efficiency" is one of the most pressing problems we encounter. Furthermore, we demonstrated an EDOF inverse design using this method. There are many possible multi-functional optics, that we can inverse design, and the reviewer may find some capabilities more interesting than others. But that is a subjective matter, and given other reviewers recommend the paper, we disagree with reviewer's opinion on this point.

We have placed our code in the public domain, and anyone can design whatever they like. We have referenced many works, and none of them have demonstrated an efficiency higher than the forward-designed lens for large aperture. Additionally, none of them demonstrated any experiment. We demonstrated a 50% maximum intensity improvement in experiment, and an experimental performance enhancement trend that matches simulation. There are no other works that do this, and we are not sure what specific additional capabilities the reviewer wants us to show. Furthermore, this is the first demonstrated performance enhancement of a metalens with a neural network approach. We believe for these reasons, and the fact that the other two reviewers have agreed that the paper demonstrates sufficient novelty, the paper should be published with no further modifications.

- they did not provide intuition for which their experiment sometimes outperforms experiments, and even indicate that their code has errors? I am not sure how to interpret that last part and the potential impact on the validity/reproducibility of the results?

Response: The reviewer is correct that we could not provide a definitive answer to explain the discrepancy between the experiment and the theory, although we observed qualitative matching. This is not the first time a scientific paper fails to explain the discrepancy, and this requires more research. The meta-optics have a large number of elements, and the random nature of fabrication errors are difficult to model. Given we are optimizing a non-convex problem, it is also possible that our theoretical design is not the global minima, and experimental errors gave us a slightly better design. Again, this is not new in the AI-assisted designs of nanophotonic structures. Without giving a made-up hypothetical reason, we clearly stated that our inability to explain results and the need of more research, which is what our scientific and ethical training taught us.

We also believe that the reviewer misinterpreted our answer on "wrong code". Our final reported results are from a correct code. However, we iterated ~7-8 designs, where our experiment did not show an enhanced efficiency for the inverse designed lens over forward-designed one. We found

that there were errors in publicly available codes that we used. Some of these papers, from where we build our code, are cited in our work already. Based on this experience, we encouraged everyone to perform experiments to validate their model (in the supplement). All these works did not perform any experiment, and possibly because of that they did not catch the errors. Again, we should treat experimental results to be accurate, and our theory qualitatively matches the experiment. We brought this up again in our response to the reviewer (and we emphasize, that this was already written in our supplement and not something we hid before) to show that there are several challenges to match the theory and experiment. Our results show experimental improvement over design only after we made sure our code is correct. "Correct" code still does not guarantee the most optimal solution for the nature of non-convex optimization in this particular case.

Overall, I do not believe that the results deserve publication in their current forms, and still encourage the authors to make significant improvements to the presentation and discussion of their results.

Response: We are sorry that the reviewer feels that way but emphasize that we have addressed the reviewer's legitimate concerns in our prior round of revision, including changes in presentation and discussion. The reviewer did not point out a distinct problem here (except misinterpretations of our statements in #2), but rather gives a generalized statement, which seems to reflect their subjective feelings, rather than a professional objective critique. Given that the two other reviewers agreed to accept the paper, we sincerely hope the editor will make a positive recommendation for the paper. The reviewer's concerns are addressed in the paper, and in our opinion, they cannot be made more satisfactory without a significant undertaking.